# Verify when Uncertain: Beyond Self-Consistency in Black Box Hallucination Detection

**Yihao Xue**                                                    *yihaoxue@g.ucla.edu*
*Department of Computer Science*
*University of California, Los Angeles*

**Kristjan Greenewald**
*MIT-IBM Watson AI Lab*

**Youssef Mroueh**
*IBM Research*

**Baharan Mirzasoleiman**
*Department of Computer Science*
*University of California, Los Angeles*

**Reviewed on OpenReview:** *https://openreview.net/forum?id=6tlLISSgiu*

## Abstract

Large Language Models (LLMs) often hallucinate, limiting their reliability in sensitive applications. In black-box settings, several self-consistency-based techniques have been proposed for hallucination detection. We empirically show that these methods perform nearly as well as a supervised (black-box) oracle, leaving limited room for further gains within this paradigm. To address this limitation, we explore cross-model consistency checking between the target model and an additional verifier LLM. With this extra information, we observe improved oracle performance compared to purely self-consistency-based methods. We then propose a budget-friendly, two-stage detection algorithm that calls the verifier model only for a subset of cases. It dynamically switches between self-consistency and cross-consistency based on an uncertainty interval of the self-consistency classifier. We provide a geometric interpretation of consistency-based hallucination detection methods through the lens of kernel mean embeddings, offering deeper theoretical insights. Extensive experiments on QA-style hallucination detection benchmarks show that this approach maintains high detection performance while significantly reducing computational cost.

## 1 Introduction

Large Language Models (LLMs) have demonstrated impressive abilities in question answering, summarization, and explanation. However, hallucinations—plausible but incorrect content–remain a persistent issue. These errors create significant challenges for both human users and tool-using or agentic applications, as the fluent presentation of false information makes verification nearly as difficult as solving the task from scratch.

A key challenge is detecting hallucinations in black-box scenarios, where we can access only an LLM's outputs but not its intermediate states—an especially relevant concern when using closed-source commercial APIs. An important intuition is that *self-consistency*—mutual entailment between multiple stochastically sampled high-temperature generations from the LLM for the same question—is lower when the model is hallucinating compared to when it provides a correct answer. As Tolstoy wrote in *Anna Karenina*, "All happy families are alike; every unhappy family is unhappy in its own way." Mechanistically, this LLM behavior can be understood as follows, from the fact that LLMs are pretrained as *statistical* language models. If the model confidently knows the answer, then probability should be concentrated on that answer. If, on the other hand,

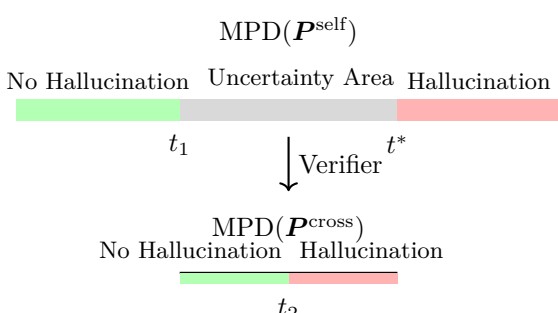

Figure 1: Two Stage Hallucination Detection. First, the self-consistency matrix $\boldsymbol{P}^{\text{self}}$ is formed and the test statistic is computed. This is thresholded with two thresholds, where medium values (gray region) advance to the second stage for disambiguation. The $\boldsymbol{P}^{\text{cross}}$ cross-consistency matrix and test statistic are then computed for these ambiguous samples for final classification.

the model does not know all or part of the answer, its statistical pretraining will bias it towards creating what is essentially a posterior distribution on answers or parts of answers that seem plausible.

Following this philosophy, prior work Manakul et al. (2023); Farquhar et al. (2024); Kuhn et al. (2023); Lin et al. (2023); Nikitin et al. (2024) has proposed various methods that leverage self-consistency. However, it remains unclear how much further they can be improved. Thus, we investigate whether we are already near the performance ceiling, given the limited information available in a black-box setting. Using a unified formalization of self-consistency-based methods, we design a method that trains graph neural networks to approximate the ceiling performance of this method family. Notably, we find that existing methods are already close to this ceiling, suggesting little room for further improvement within this paradigm.

This highlights the need to go beyond self-consistency alone. Thus, we consider the case where an additional model serves as a verifier, and we incorporate consistency checking between answers generated by the target model and the verifier. This approach provides information that self-consistency alone cannot capture. For example, agreement between the two models increases confidence in a response's correctness, while disagreement suggests that at least one model is likely hallucinating. Through experiments, we observe a significant gain in the approximated ceiling performance when both self-consistency and cross-model consistency are taken into account, interestingly, even when the verifier is weaker than the target model. Additionally, we find that linearly combining self-consistency with cross-model consistency can achieve performance very close to this ceiling.

Finally, we address the computational overhead introduced by the verifier model. We propose a budget-aware method that performs cross-model consistency checking for only a specified fraction of questions, keeping the computation budget controllable. This method consists of two stages (illustrated in Fig. 1): first, it performs self-consistency checking; then, it selectively applies cross-model consistency checking only when self-consistency falls in a middle range, where judgment is less reliable. We provide a geometric interpretation of this approach through the lens of kernel mean embeddings, offering theoretical insights into its effectiveness. Through extensive experiments across three datasets and 20 target-verifier combinations, we demonstrate that this adaptive mechanism can achieve high detection performance while significantly reducing computational costs. Additionally, we provide practical suggestions on selecting verifier models based on different budget constraints.

We note that this work focuses on hallucination detection in QA settings, following prior work on black-box consistency-based methods. While recent benchmarks have begun to evaluate hallucinations in other generation tasks such as summarization and long-form generation, evaluation in these settings is generally more challenging, as it is harder to compare a model's output with a ground-truth answer when both are long and open-ended. Determining whether two long passages are semantically consistent or whether a specific detail constitutes a hallucination is inherently ambiguous. Some evaluation, e.g., that of Manakul et al. (2023), even requires manual annotation of hallucinations, which can be expensive, time-consuming, and subjective. In contrast, QA has a more well-defined target. Therefore, we focus on QA-style benchmarks as a standard and controlled setting for studying consistency-based signals.

## 2 Related Work

There are works that explore white-box detection methods, such as Duan et al. (2024); Varshney et al. (2023), which require token-level logits, or Yin et al. (2024); Zou et al. (2023); Agrawal et al. (2023), which rely on intermediate representations. White-box methods are less suitable for certain scenarios, such as closed-source commercial APIs. In this work, we focus exclusively on black-box hallucination detection, where we do not have access to the internal workings of the LLM. In this scenario, the primary approach involves checking the consistency between multiple samples of the LLM's answers Manakul et al. (2023); Farquhar et al. (2024); Kuhn et al. (2023); Lin et al. (2023); Nikitin et al. (2024). These works rely on sampling multiple answers to the same question from the LLM and using an NLI (Natural Language Inference) model to determine whether they are semantically equivalent. The NLI judgments are then processed in various ways to decide whether a hallucination has occurred. Details of these methods are discussed in Section 4. Manakul et al. (2023); Kuhn et al. (2023) also explore alternative methods for judging semantic equivalence, which are either less effective (e.g., n-gram) or computationally expensive (e.g., using an LLM). Zhang et al. (2023) identify limitations in self-consistency-based methods and propose leveraging question perturbation and cross-model response consistency (comparing an LLM's responses with those from an additional verifier LLM) to improve performance. While their approach improves results, introducing a verifier model adds computational overhead. In this work, we systematically explore the possibility of achieving computational efficiency when combining self-consistency and cross-model consistency. Note that the question perturbation technique from Zhang et al. (2023) is orthogonal to our approach and could potentially be incorporated to achieve better results. Another line of work involves directly asking LLMs to judge the uncertainty of their answers Mielke et al. (2022); Tian et al. (2023b); Kadavath et al. (2022); Lin et al. (2022), which typically requires additional finetuning/calibration and does not fit within the black-box scenario. Without any modification to the original LLMs, their verbalized confidence is often inaccurate Xiong et al. (2023). The inherent conflict between calibration and hallucination, as theoretically shown in Kalai & Vempala (2024), further highlights the limitations of this approach. There are also works addressing hallucination mitigation, such as using RAG Asai et al. (2023); Gao et al. (2022), inference-time intervention Li et al. (2024), or fine-tuning Lee et al. (2022); Tian et al. (2023a), which is beyond the scope of this paper.

## 3 Preliminaries

In the task of hallucination detection, we have a target LLM, denoted by $\mathcal{M}_t$, for which we aim to detect hallucinations. We are given a set of questions $\{q_i\}_{i=1}^n$. Given a set of questions $\{q_i\}_{i=1}^n$, the model generates answers, $a_i = \mathcal{M}_t(q_i, \tau)$, under a specified temperature $\tau$. The ground truth annotation $\hat{h}_i$ indicates whether $a_i$ is a hallucination ($\hat{h}_i = 1$) or factual ($\hat{h}_i = 0$). The objective is to predict whether $a_i$ is a hallucination, with our prediction denoted by $h_i$.

To achieve this, many methods are designed to output a value, $v_i$, that captures specific characteristics (e.g., the uncertainty of the answer). A higher value of $v_i$ suggests that $a_i$ is more likely to be a hallucination. The prediction $h_i$ is then determined based on a threshold applied to $v_i$, where the choice of threshold dictates the final classification.

To evaluate the performance of a hallucination detection method, we focus on two widely accepted metrics computed given outputs $\{v_i\}_{i=1}^n$ and ground truths $\{\hat{h}_i\}_{i=1}^n$: (1) **AUROC**, area under the receiver operating characteristic curve. is a classic performance measure in binary classification. It captures the trade-off between the true positive rate and the false positive rate across various thresholds, providing an aggregate measure of the model's ability to distinguish between the two classes. (2) **AURAC**, the area under the "rejection accuracy" curve Farquhar et al. (2024). It is designed for scenarios where a hallucination detection method is employed to refuse answering questions that the model is most likely to hallucinate on. Rejection accuracy measures the model's accuracy on the $X\%$ of questions with the lowest $v_i$ values (least likely to hallucinate), and the area under this curve summarizes performance across all values of $X$.

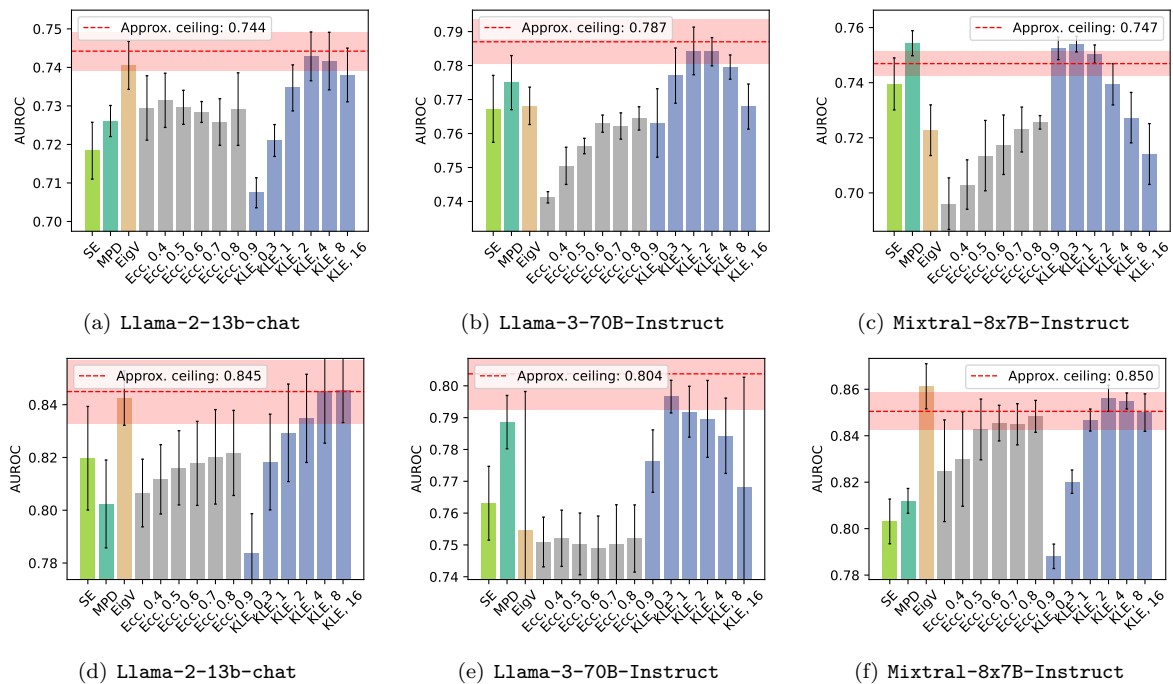

Figure 2: Comparison between AUROC of existing methods and the approximated ceiling performance on SQuAD ((a)–(c)) and TriviaQA ((d)–(f)). The best method performs very close to the oracle, indicating that we are approaching the performance limit. Similar results for AURAC are in Fig. 7.

## 4 From Self-consistency to Cross-Consistency

### 4.1 Self-consistency based detection

Prior work Manakul et al. (2023); Farquhar et al. (2024); Kuhn et al. (2023); Lin et al. (2023); Nikitin et al. (2024) has introduced various methods leveraging self-consistency. However, the extent to which these methods can be further improved remains unclear. To explore this, we develop a method to approximate the ceiling performance for any approach that utilizes self-consistency and compare it against the performance of existing methods.

**Unified formalization of self-consistency-based methods.** We first present a unified formalization of the existing methods. Recall that the goal is to determine whether each $\mathcal{M}(q_i)$ is a hallucination. All these methods rely on additionally sampling $m$ answers from the LLM $\mathcal{M}$ for question $q_i$ under a high temperature $\tau'$, which is typically much higher than $\tau$, the temperature used to generate $a_i$. For example, in Farquhar et al. (2024), the settings are $\tau = 0.1$ and $\tau' = 1.0$. Let $\{a'_{i,j}\}_{j=1}^m$ denote the set of these additionally sampled answers. These methods then use an entailment estimator (e.g., deberta-v2-xlarge-mnli), denoted by $\mathcal{E}$. The estimator $\mathcal{E}$ takes two answers as input and outputs a value between 0 and 1, indicating the degree of entailment between the two answers, where 1 means full entailment. Using $\mathcal{E}$, a self-entailment matrix $\boldsymbol{P}_i^{\text{self}}$ is constructed as: $\boldsymbol{P}_i^{\text{self}} = [\mathcal{E}(a'_{i,j}, a'_{i,k})]_{1 \le j \le m, 1 \le k \le m}$, where each element is the entailment value for a pair of answers in $\{a'_{i,j}\}_{j=1}^m$. Existing methods can then be formalized as some function $f$ applied to the self-entailment matrix $\boldsymbol{P}_i^{\text{self}}$ which outputs a scalar. The focus of prior work lies in designing various forms of $f$. Specifically, (1) SE($\boldsymbol{P}^{\text{self}}$), the Semantic Entropy Farquhar et al. (2024), uses a binarized version of $\boldsymbol{P}_i^{\text{self}}$ to identify which answers belong to the same semantic set and then computes the entropy over these semantic sets. (2) MPD($\boldsymbol{P}_i^{\text{self}}$) Lin et al. (2023); Manakul et al. (2023) is simply the mean pairwise distance, computed as $1 - \text{Mean}(\boldsymbol{P}_i^{\text{self}})$. (3) EigV($\boldsymbol{P}_i^{\text{self}}$) Lin et al. (2023) is defined as the sum of the eigenvalues of the graph Laplacian of $\boldsymbol{P}_i^{\text{self}}$. (4) Ecc($\boldsymbol{P}_i^{\text{self}}$) Lin et al. (2023) measures the eccentricity of the answers leveraging the eigenvectors of the graph Laplacian of $\boldsymbol{P}_i^{\text{self}}$. (5) KLE($\boldsymbol{P}_i^{\text{self}}$), the Kernel Language Entropy Nikitin et al. (2024), involves applying the von Neumann entropy to a graph kernel derived from $\boldsymbol{P}_i^{\text{self}}$. For all these methods, a higher output indicates greater uncertainty among $\{a'_{i,j}\}_{j=1}^m$, making the corresponding low-temperature answer $a_i$ more likely to be a hallucination.

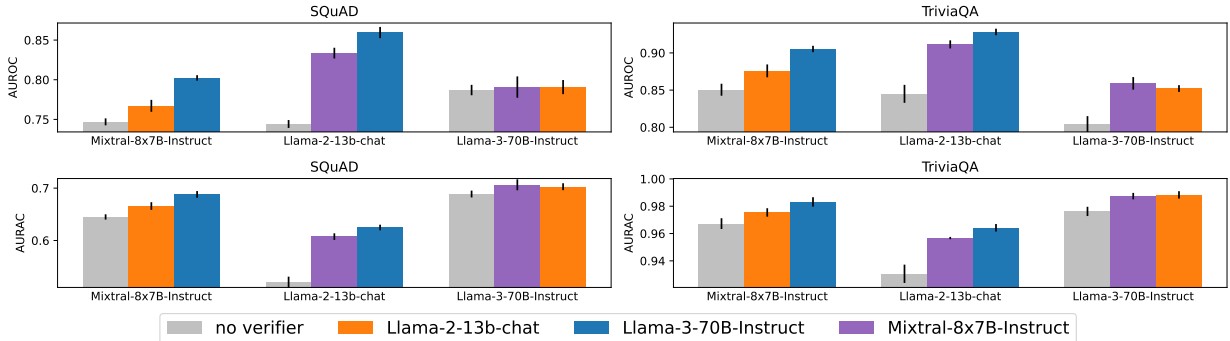

Figure 3: Comparison between approximated ceiling performances using only $\boldsymbol{P}^{\mathrm{self}}$ (gray) and those using both $\boldsymbol{P}^{\mathrm{self}}$ and $\boldsymbol{P}^{\mathrm{cross}}$. The x-axis shows the target model, and the colors indicate the verifier model, as shown in the legend. We observe a clear improvement when a verifier model is used.

The underlying assumption is that $\boldsymbol{P}_i^{\mathrm{self}}$ contains exploitable information related to $\hat{h}_i$, the ground truth hallucination annotation. This prompts the question: how much information does $\boldsymbol{P}^{\mathrm{self}}$ actually encode about $\hat{h}_i$? To explore this, we aim to identify the optimal function $f$ that maps $\boldsymbol{P}^{\mathrm{self}}$ to the hallucination label. This leads to the following formulation:

$$\hat{f} = \arg\min_{f} \mathbb{E}[l(f(\boldsymbol{P}^{\mathrm{self}}), \hat{h})], \tag{1}$$

where $l$ is a loss function that measures the discrepancy between the output value and the actual label.

**Approximating the ceiling performance with GCN models.** To search for $\hat{f}$, we frame it as a learning problem. Since the task is ultimately binary classification based on the matrix $\boldsymbol{P}^{\mathrm{self}}$, graph neural networks are well-suited due to their ability to process matrix structures and express a wide range of functions. We use a two-layer Graph Convolutional Network (GCN) to represent $f$. The model is trained with BCE loss on sampled pairs of $\boldsymbol{P}^{\mathrm{self}}$ and $\hat{h}$. We then evaluate AUROC and AURAC of the resulting model as an approximation of the ceiling performance. The training and test samples are drawn independently to account for the finiteness of the data, ensuring that the evaluation reflects the model's ability of capturing a generalizable relationship between $\boldsymbol{P}^{\mathrm{self}}$ and $\hat{h}$, rather than that of overfitting the training data.

**Results.** In Figs. 2 and 7, we compare the performance of existing methods with the ceiling performance approximated using the aforementioned approach across various settings. We consider three different LLMs: `Llama-2-13b-chat`, `Llama-3-70B-Instruct`, and `Mixtral-8x7B-Instruct-v0.1`, as well as two datasets: SQuAD and TriviaQA. In the plots, the x-axis represents different methods with varying hyperparameters. The best-performing method varies across different settings, with MPD, KLE, and EigV consistently showing relatively strong performance. Notably, in each setting, the top method closely approaches the approximated ceiling, indicating that existing methods already make near-maximal use of $\boldsymbol{P}^{\mathrm{self}}$, particularly when sufficient validation data is available to optimize method and hyperparameter selection.

## 4.2 Incorporating cross-model consistency

As noted in the previous subsection, existing methods bring us very close to the ceiling performance for self-consistency alone. The question now is: how can we push beyond this limit? In the black-box scenario, our options are constrained by the lack of access to any internal model information. Zhang et al. (2023) has explored another potential approach: leveraging outputs from other LLMs to improve hallucination detection through cross-model comparisons. A minimal case involves using one additional model as a verifier. This added layer of information can help further refine hallucination detection. For example, if two models significantly disagree on their answers, at least one is likely hallucinating.

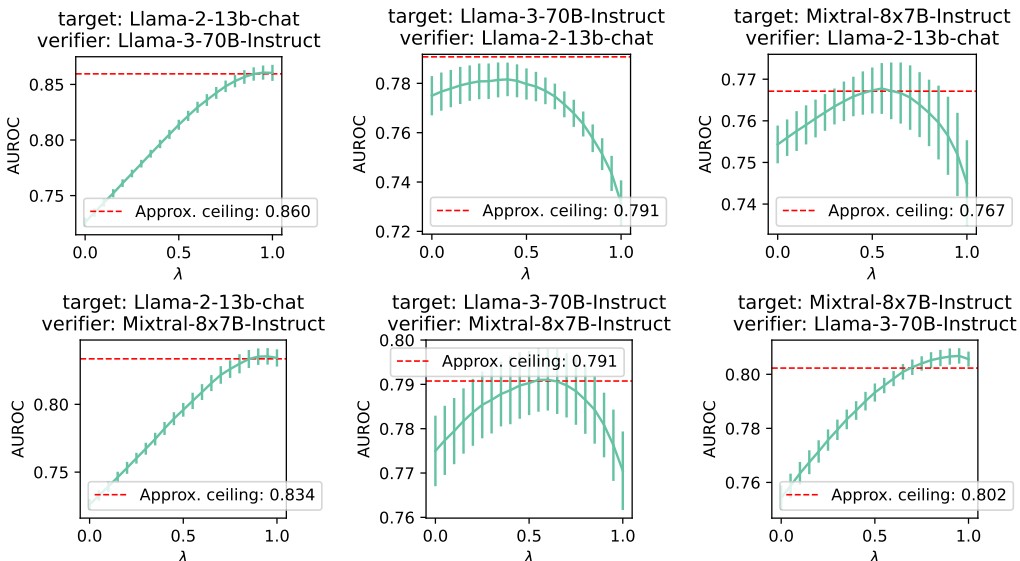

Figure 4: A simple weighted average of self-consistency and cross-consistency-based metrics, $(1 - \lambda)\mathrm{MPD}(\boldsymbol{P}^{\mathrm{self}}) + \lambda\mathrm{MPD}(\boldsymbol{P}^{\mathrm{cross}})$, can achieve performance close to that of the oracle method. Plots for AURAC are in Fig. 8 in Appx. C.2.

### 4.2.1 Improvement in the ceiling performance

We explore how much gain cross-model consistency checking can possibly bring. We denote the verifier model as $\mathcal{M}_v$. Similar to the self-consistency case, a natural extension is to encode the cross-model consistency information in a matrix: $\boldsymbol{P}_i^{\mathrm{cross}} = [\mathcal{E}(a'_{i,j}, b'_{i,k})]_{1 \le j \le m, 1 \le k \le m}$, where $\{b'_{i,k}\}_{k=1}^m$ are $m$ answers sampled from $\mathcal{M}_v$ under temperature $\tau'$ for question $q_i$. Thus, $\boldsymbol{P}_i^{\mathrm{cross}}$ captures the pairwise entailment relationships between the answers generated by the target $\mathcal{M}_t$ and the verifier $\mathcal{M}_v$.

*Remark* 4.1 (Cross entailment). To build intuition, consider the setting of a very strong verifier model that always returns a sample from the ground truth. Then, if the entailment model returns a calibrated posterior probability of entailment, it is easy to see that $\mathrm{Mean}(\boldsymbol{P}_i^{\mathrm{cross}})$ is the probability of entailment between an $\mathcal{M}_t$ sample and a ground truth sample. In other words, it can be interpreted as the probability of correctness. As the verifier weakens, we hypothesize that the $\mathrm{Mean}(\boldsymbol{P}_i^{\mathrm{cross}})$ retains significant correlation with the probability of correctness, and observe this in practice.

Building on the formalization in Section 4.1, we aim to determine how much information can be extracted when both $\boldsymbol{P}^{\mathrm{self}}$ and $\boldsymbol{P}^{\mathrm{cross}}$ are used to predict the ground truth hallucination label. To achieve this, we search for a function $f$ that takes both $\boldsymbol{P}^{\mathrm{self}}$ and $\boldsymbol{P}^{\mathrm{cross}}$ as input. Given the pairwise nature of the data, we again leverage GCN models to represent the function. Specifically, we combine $\boldsymbol{P}^{\mathrm{self}}$ and $\boldsymbol{P}^{\mathrm{cross}}$ into a single matrix: $\begin{bmatrix} \boldsymbol{P}^{\mathrm{self}} & \boldsymbol{P}^{\mathrm{cross}} \\ \boldsymbol{P}^{\mathrm{cross}} & \boldsymbol{0} \end{bmatrix}$ which encodes the underlying structure of the data as pairwise relationships between answers. We then apply a GCN to this combined matrix and train the model to fit the ground truth labels $\hat{h}$. Finally, we evaluate the resulting model to approximate the ceiling performance achievable with both $\boldsymbol{P}^{\mathrm{self}}$ and $\boldsymbol{P}^{\mathrm{cross}}$.

We now compare the approximated ceiling performances achieved using only $\boldsymbol{P}^{\mathrm{self}}$ to those achieved using both $\boldsymbol{P}^{\mathrm{self}}$ and $\boldsymbol{P}^{\mathrm{cross}}$ in Fig. 3. The x-axis represents the target model, and the colors indicate the verifier model used. Gray bars correspond to the scenario where only $\boldsymbol{P}^{\mathrm{self}}$ is used, i.e., no verifier model is involved. The results demonstrate a clear improvement in performance, measured by both AUROC and AURAC, when a verifier model is introduced. Interestingly, this improvement is observed even in cases where the target model itself is quite strong. For example, on TriviaQA, adding a weaker verifier model can still enhance detection performance when `Llama-3-70B-Instruct` is used as the target model. This highlights

---

**Algorithm 1** Budget-aware two-stage detection

---

**Input:** question $q$, target model $\mathcal{M}_t$, verifier model $\mathcal{M}_v$, number of samples $m$, temperature $\tau'$, entailment estimator $\mathcal{E}$, thresholds $t_1, t_2, t^*$.

$\{a'_j\}_{j=1}^m \leftarrow$ the $m$ responses sampled from $\mathcal{M}_t$ for the question $q$ under temperature $\tau'$         ▷ First stage

$\boldsymbol{P}^{\text{self}} \leftarrow [\mathcal{E}(a'_j, a'_k)]_{1 \le j \le m, 1 \le k \le m}$ and $s^{\text{self}} \leftarrow \text{MPD}(\boldsymbol{P}^{\text{self}})$

**if** $s^{\text{self}} < t_1$ **then return** $h = $ **False**         ▷ no hallucination

**else if** $s^{\text{self}} > t^*$ **then return** $h = $ **True**         ▷ hallucination

**else**

    $\{b'_j\}_{j=1}^m \leftarrow$ the $m$ responses sampled from $\mathcal{M}_v$ for the question $q_i$ under temperature $\tau'$         ▷ Second stage

    $\boldsymbol{P}^{\text{cross}} \leftarrow [\mathcal{E}(a'_j, b'_k)]_{1 \le j \le m, 1 \le k \le m}$ and $s^{\text{cross}} \leftarrow \text{MPD}(\boldsymbol{P}^{\text{cross}})$ **return** $h = $ **False** if $s_i^{\text{cross}} < t_2$ else **True**

**end if**

---

the potential of leveraging cross-model consistency, as even a less powerful verifier can provide complementary insights that enhance hallucination detection.

### 4.2.2 Linearly combining MDPs closely approaches the ceiling

Although the function the GCN implements to achieve ceiling performance is unknown, interestingly, we find that a simple extension of existing methods can perform almost equally well. We leverage MPD introduced earlier, which can be naturally extended to $\boldsymbol{P}^{\text{cross}}$ as $\text{MPD}(\boldsymbol{P}^{\text{cross}}) = 1 - \text{Mean}(\boldsymbol{P}^{\text{cross}})$. The combined approach uses a weighted average: $(1 - \lambda)\text{MPD}(\boldsymbol{P}^{\text{self}}) + \lambda\text{MPD}(\boldsymbol{P}^{\text{cross}})$, where $\lambda$ is a hyperparameter. As shown in Fig. 4, with an appropriate choice of $\lambda$, this method achieves performance very close to the approximated ceiling performance.

## 5 Budget-Aware Hallucination Detection with A Verifier Model

From the previous section, we observe that performance improves significantly when self-consistency checking is combined with cross-model consistency checking. However, this approach can introduce substantial computational overhead, especially when the verifier is a large model. For instance, with a 7B target model and a 70B verifier model, cross-model consistency adds 10 times the computation.

To address this issue, we propose a method to control computational overhead. As illustrated in Fig. 1. The key idea is to perform cross-model consistency checking only when it is most necessary. Intuitively, when self-consistency scores are extremely high or low, it is likely—though not guaranteed—that the model's output is non-hallucinatory or hallucinatory, respectively. In such cases, performing cross-model consistency checking may not be necessary under limited computational budgets. Instead, cross-model consistency should focus on cases with intermediate self-consistency scores, where our judgment is more uncertain. This is formalized in Alg. 1. The parameter $p$ specifies the fraction of instances for which cross-model consistency checking will be performed. The parameter $t_1$ is the threshold for $\text{MPD}(\boldsymbol{P}^{\text{self}})$ below which the output is deemed non-hallucinatory. Based on $t_1$, we choose to compute $t^*$, the threshold for $\text{MPD}(\boldsymbol{P}^{\text{self}})$ above which the output is classified as hallucinatory, such that only $p$ of the $\text{MPD}(\boldsymbol{P}^{\text{self}})$ scores fall between $t_1$ and $t^*$. For these intermediate cases, judgments are made based on cross-model consistency $\text{MPD}(\boldsymbol{P}^{\text{cross}})$ using a threshold $t_2$.[1] Note that we use MPD to measure inconsistency from $\boldsymbol{P}^{\text{self}}$ due to its simplicity, ease of extension to $\boldsymbol{P}^{\text{cross}}$ (unlike, e.g., KLE, which is specifically designed for $\boldsymbol{P}^{\text{self}}$ but not immediately well-defined for $\boldsymbol{P}^{\text{cross}}$), and the fact that it contains sufficient information for achieving ceiling performance, as discussed in Sec. 4.2.2. Future work could explore other metrics.

**Geometric Interpretation in Mean Embedding Space.** We provide a geometric interpretation of our hallucination detection. For each prompt $x$ we can observe the conditional distribution of the target model $\pi_t(y|x)$ and the verifier $\pi_v(y|x)$. In particular we observe $\frac{1}{n_a}\sum_{i=1}^{n_a}\delta_{y_i^a}$, $y_i^a \sim \pi_t(.|x)$ and $\frac{1}{n_v}\sum_{i=1}^{n_v}\delta_{y_i^v}, y_i^v \sim \pi_v(.|x)$. We assume that the symmetrized[2] entailment kernel $\mathcal{E}' : \mathcal{Y} \times \mathcal{Y} \to [0, 1]$ to be a reproducing kernel. The mean embeddings Muandet et al. (2017) of target, verifier and ground truth

---

[1]We threshold $\text{MPD}(\boldsymbol{P}^{\text{cross}})$ directly in the second stage (instead of a linear combination as in Sec. 4.2.2, since (a) it saves a hyperparameter and (b) we are able to approach the GCN-based performance ceiling without it.

[2] A common strategy adopted in prior work Manakul et al. (2023); Farquhar et al. (2024); Nikitin et al. (2024); Kuhn et al. (2023); Lin et al. (2023) is to construct a symmetric entailment matrix by averaging entailment scores in both directions—e.g.,

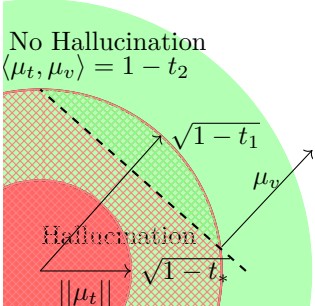

Figure 5: Geometric interpretation in mean embedding spaces of "target" ($\mu_t$) and "verifier" distributions ($\mu_v$). Self-consistency is measured via the norm of the target model's mean embedding, and cross-consistency via the dot product between mean embeddings. Stage one detects hallucination using $\|\mu_t\|$, with no hallucination outside the green sphere of radius $\sqrt{1-t_1}$ and hallucination inside the red sphere of radius $\sqrt{1-t_*}$. Between these, a hyperplane defined by $\mu_v$ and $t_2$ separates hallucination (dashed red) and no hallucination (dashed green) zones.

are respectively $\mu_t = \frac{1}{n_a} \sum_{i=1}^{n_a} \mathcal{E}(y_i^a, .)$, $\mu_v = \frac{1}{n_v} \sum_{i=1}^{n_v} \mathcal{E}'(y_i^v, .)$, and $\mu^* = \frac{1}{n^*} \sum_{i=1}^{n^*} \mathcal{E}'(y_i^*, .)$ We can write the self-consistency in terms of norms of mean embeddings : $\|\mu_t\|^2 = \frac{1}{n_a^2} \sum_{i,j} \mathcal{E}'(y_i^a, y_j^a) = 1 - \mathrm{MPD}(\boldsymbol{P}^{\text{self}})$ and the cross-consistency $\langle \mu_t, \mu_v \rangle = \frac{1}{n_a n_v} \sum_{i,j} \mathcal{E}'(y_i^a, y_j^v) = 1 - \mathrm{MPD}(\boldsymbol{P}^{\text{cross}})$. Fig. 5 gives a geometric interpretation of our two stage algorithm in means embedding spaces. If the "target" model norm of mean embedding is higher than a threshold $\sqrt{1-t_1}$ no hallucination is detected and for a norm less than a threshold $\sqrt{1-t^*}$ is detected. For the uncertainty area between the two spheres, the verifier mean embedding defines with the threshold $1 - t_2$ a hyperplane that divides this area in no hallucination (above) and hallucination (below).

**Evaluating AUROC/AURAC for Algorithm 1.** Recall that, in the conventional scenario, the AUROC and AURAC metrics are defined for a system that outputs a single value for binary classification. To compute these metrics, we vary the threshold used to produce the label from the output. In such cases, for both ROC and RAC curves, each $X$-value corresponds to a single $Y$-value. For example, in ROC, one false positive rate corresponds to one true positive rate, allowing us to obtain a single curve by varying the threshold and then compute the area under it. However, the situation is more complex for our Algorithm 1, which, given a fixed hyperparameter $p$, ultimately outputs binary labels but involves two thresholds, $t_1$ and $t_2$. Different combinations of $t_1$ and $t_2$ can yield the same $X$-value but different $Y$-values, resulting in a plot that resembles a thick band rather than a single curve. Thus, the area under the curve is not well-defined. The way these thresholds are adjusted relative to each other significantly affects the $Y$ values for a given $X$. To derive a meaningful performance measure for our algorithm, we establish the relationship between the two thresholds using a validation set. Specifically, on this set, we iterate over a grid of threshold combinations, running the algorithm to obtain all $X, Y$ pairs. For each small interval of $X$, we identify the threshold combinations that maximize $Y$. Next, we use a separate test set, where we only evaluate the good combinations identified from the validation step and compute the resulting AUROC/AURAC. We note that prior work Lin et al. (2023); Nikitin et al. (2024) also relies on a validation set for tuning hyperparameters.

We now prove that the above procedure for selecting the two thresholds from data achieves a test-time AUROC close to the optimal value achieved on the validation set. Our theorem below applies more generally to any method that uses data to select from a finite set of threshold values.

**Theorem 5.1** (AUROC Generalization)*. Suppose we are given $n_{neg}$ i.i.d. samples from the non-hallucinating distribution and $n_{pos}$ i.i.d. samples from the hallucinating distribution, and sets of candidate thresholds $\mathcal{T}_1 = \{t_j^1\}_{j=1}^{|\mathcal{T}_1|}$ and $\mathcal{T}_2 = \{t_k^2\}_{k=1}^{|\mathcal{T}_2|}$ for stages 1 and 2 respectively. Suppose we use this data to choose a mapping $t_1, t_2 = \mathcal{A}(p_{FA})$ from desired probability of false alarm level $p_{FA} \in [0,1]$ to thresholds $t_1 \in \mathcal{T}_1, t_2 \in \mathcal{T}_2$, maximizing the probability of detection on the validation data. Let $A_{val}(\mathcal{A})$ be the AUROC using thresholds given by $\mathcal{A}$. Then, with probability at least $\left(1 - \frac{2}{|\mathcal{T}_1||\mathcal{T}_2|}\right)^2$, the test AUROC satisfies $A_{test}(\mathcal{A}) \geq A_{val}(\mathcal{A}) - 2\epsilon$ and the test $|p_{FA}^{test} - p_{FA}| \leq \epsilon$, where $\epsilon = \sqrt{\frac{\log(|\mathcal{T}_1|)+\log(|\mathcal{T}_2|)}{\min(n_{neg}, n_{pos})}}$.*

See proof in App. A. This theorem implies we need to have $n_{neg}, n_{pos} = \Omega\left(\log(|\mathcal{T}_1|) + \log(|\mathcal{T}_2|)\right)$ to guarantee the test AUROC is close to the convex-hull-AUROC on validation data. In Section 6, we validate through experiments that the selected thresholds transfer well to the test data.

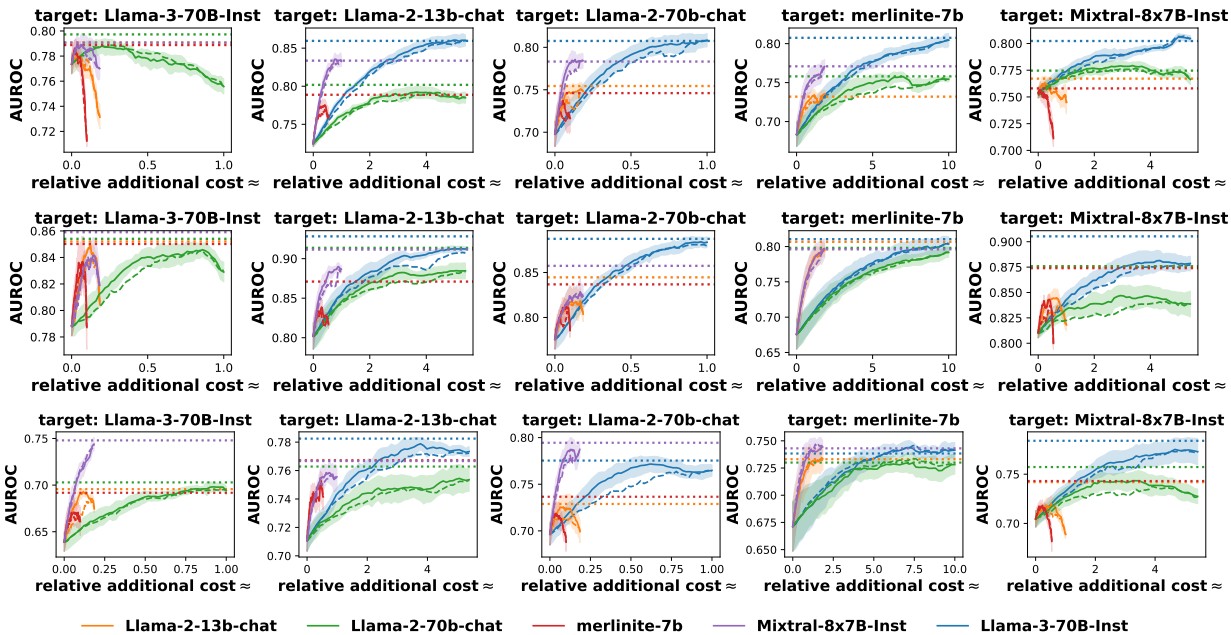

Figure 6: We plot AUROC against the *relative additional cost* for SQuAD (top), TriviaQA (middle), and NQ (bottom). Solid curves represent results where the validation set consists of independent samples for the same questions, while dashed curves correspond to validation sets consisting of answers for different questions. The dotted horizontal line indicates the approximate ceiling performance using GNN. The curves are mostly convex, indicating that a small number of cross-model consistency checks contribute to most of the performance gain. This demonstrates that our approach can achieve high performance with very low cost, further evidenced in Table 1. Results for AURAC are in Fig. 9.

## 6 Experiments

**Datasets.** We consider datasets widely used in research on hallucination detection Kuhn et al. (2023); Farquhar et al. (2024); Lin et al. (2023); Nikitin et al. (2024): TriviaQA Joshi et al. (2017) for trivia knowledge, SQuAD Rajpurkar et al. (2016) for general knowledge, and Natural Questions Kwiatkowski et al. (2019), derived from real user queries to Google Search.

**Models.** We consider: `Llama-3-70B-Instruct`, `Llama-2-70b-chat-hf` `Llama-2-13b-chat`, `Mixtral-8x7B-Instruct-v0.1` and `merlinite-7b`, resulting in 20 target-verifier pairs. Following prior works Lin et al. (2023); Kuhn et al. (2023); Nikitin et al. (2024), we use `deberta-v2-xlarge-mnli` as the entailment estimator, taking the post-softmax probability of the 'Entailment' label as the output (ranging from 0 to 1).

**Evaluation.** We set $\tau = 0.1, \tau' = 1.0, m = 10$ [3]. To obtain the ground truth annotations for hallucination, we use GPT-4 as the judge [4]. Performance is measured in terms of AUROC and AURAC as described in Section 6. Since this requires a validation set, we compare two scenarios: (1) the validation set consists of the same questions as the test set but with independently sampled random answers, and (2) the validation set consists of different questions from the same dataset. In both scenarios, the sizes of the validation and test sets are 400. First scenario represents an oracle setting, as the thresholds rely on the ground truth for the

---

using $0.5\mathcal{E}(a'_{i,j}, a'_{i,k}) + 0.5\mathcal{E}(a'_{i,k}, a'_{i,j})$ for both entries $(j, k)$ and $(k, j)$. Note that, for MPD, using either the asymmetric or symmetric version yields the same result.

[3]We note that $m = 10$ is the most common choice in the literature Manakul et al. (2023); Farquhar et al. (2024); Kuhn et al. (2023); Lin et al. (2023); Nikitin et al. (2024). As noted in Lin et al. (2023), while performance generally improves as $m$ increases, the gain plateaus once $m$ exceeds a small value (e.g., between 3 and 5, as shown in their Figure 4). Additionally, increasing $m$ also raises the computational cost. Therefore, we fix $m = 10$ in our experiments.

[4]An analysis of GPT-4's use in labeling hallucinations, compared with human annotators, can be found in Farquhar et al. (2024). Specifically, in the supplementary material ("Note 6: Assessing Model Accuracy"), the authors show that two human raters agreed with each other at roughly the same rate (92%) as they agreed with GPT-4 on average (93%). This suggests that GPT-4's evaluations are, on average, reasonably close to those of human raters.

Table 1: The minimum $p$ (percentage of verifier calls) required to achieve $\alpha\%$ of the maximal AUROC gain (denoted as $p(\alpha)$) when `Llama-3-70B-Instruct` is the verifier, for different values of $\alpha$. We report the average results across all target models on SQuAD (S), TriviaQA (T) and Natural Questions (N). The last column shows the maximal AUROC gain ($\Delta_{\max}$).

|   | $p(70)$ | $p(80)$ | $p(90)$ | $p(95)$ | $\Delta_{\max}$ |
|---|---|---|---|---|---|
| S | $48.5_{\pm 7}$ | $62.5_{\pm 11}$ | $83.0_{\pm 5}$ | $86.5_{\pm 5}$ | $0.1_{\pm 0.03}$ |
| T | $43.0_{\pm 3}$ | $53.0_{\pm 2}$ | $70.5_{\pm 8}$ | $78.5_{\pm 10}$ | $0.1_{\pm 0.02}$ |
| N | $39.5_{\pm 6}$ | $52.0_{\pm 10}$ | $65.0_{\pm 6}$ | $72.0_{\pm 6}$ | $0.07_{\pm 0.00}$ |

same questions. The second scenario reflects a practical setting where a separate set of questions is used to determine thresholds. We repeat the test with 5 random seeds and report the average result along with the standard deviation.

**Estimation of computational overhead.** Given that Alg. 1 focuses on budget awareness, it is important to consider both performance and budget in our evaluation. To quantify the additional computational cost introduced in Alg. 1 compared to the case where only self-consistency is used, we define a metric called *relative additional cost*:

$$\frac{\text{FLOPs(Alg. 1)} - \text{FLOPs(only self-consistency checking)}}{\text{FLOPs(only self-consistency checking)}},$$

which can be estimated as $\frac{pN_v}{N_t}$ using the formula from Kaplan et al. (2020) (detailed derivation in App. C.1), $N_t$ and $N_v$ represent the number of non-embedding parameters in the target and verifier models, respectively. $p$ accounts for the fact that the verifier model is queried for only a fraction $p$ of the questions in Alg. 1.

**Performance vs. cost.** In Figs. 6 (AUROC) and 9 (AURAC) (App. C.2), we plot the detection performance against the estimated *relative additional cost* when varying $p$. A general trend is that when the verifier model is stronger than the target model (e.g., when the target is `merlinite-7b` and the verifier is `Llama-3-70B-Instruct`), increasing the computational budget $p$—by allowing more verifier calls—monotonically improves performance. However, with a weaker verifier (e.g., when the target is `Llama-3-70B-Instruct` and the verifier is `Llama-2-13b-chat`), we observe an increasing-decreasing trend, where an intermediate number of verifier calls achieves the best results. This aligns with the intuition from Fig. 4, where intermediate weights on self-consistency cross-model-consistency are optimal, indicating that even a weaker verifier can contribute meaningfully to detection when an appropriate balance is maintained. We include an example in App. C.2 showing a case where the output of the weak verifier is better than that of the target model. In all cases, our method—which incorporates an additional verifier—outperforms the self-consistency-only baselines (i.e., the point at "relative additional cost = 0" in Fig. 6, and other self-consistency-based methods shown in Fig. 2).

**Our approach can achieve high performance with very low computational cost.** The curves in Fig. 6 are convex, indicating that a small number of cross-model consistency checks contributing to most of the improvement in performance. To further illustrate this, we examine the minimum $p$ required to achieve different percentages of performance gains when using `Llama-3-70B-Instruct` as the verifier in Table 1. Notably, compared to querying the verifier for all questions, we can reduce the cost by 13.5%–28% while retaining 95% of the gain ($\Delta_{\max}$) in performance.

**Selection of the verifier.** `Llama-3-70B-Instruct` (blue), consistently achieves the best results when the computational budget is large. However, `Mixtral-8x7B-Instruct-v0.1` (purple), stands out for its exceptional performance with a very small cost. This efficiency can be attributed to its MoE based design—despite having 46.7B total parameters, it only uses 12.9B parameters per token.

**Transferability of thresholds.** The gap between the scenarios where the validation set contains the same questions or different questions is overall small (dashed vs. solid lines, Fig 6), suggesting that the selection of thresholds transfers well to different questions, making the approach practical.

**Comparison with approximated ceiling performance.** The gap between our approach and the ceiling performance approximated using supervised learning with GCNs (described in Sec. 4.2.1; horizontal dashed

line, Fig 6) is generally small. This indicates that our method effectively utilizes the verifier's information despite not always querying it.

## 7 Conclusion and Discussion

In this paper, we empirically analyzed consistency-based methods for hallucination detection. Based on this analysis, we introduced a budget-aware two-stage approach that leverages both self-consistency and cross-model consistency with a given verifier. Our method reduces computational cost by selectively querying the verifier. Extensive experiments demonstrate that it achieves strong performance with minimal computation, notably reducing computation by up to 28% while retaining 95% of the maximal performance gain. One limitation is that our approach currently requires a validation set to determine thresholds; future work may explore ways to remove this dependency.

While our experiments are conducted on QA datasets, the proposed framework itself is task-agnostic. Both self-consistency and cross-model consistency operate purely on distributions of semantic entailment judgments. The ceiling analysis and the two-stage detection method do not rely on properties unique to QA, but instead on the distribution of entailment relations among sampled responses. As a result, we expect similar saturation phenomena and verifier-based gains to arise in other generation settings where semantic equivalence can be assessed.

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

# A Proof of Theorem 5.1

A classifier $C$ maps points to binary predictions. The $p_D(C)$ is the probability that positive samples are mapped to 1, and $p_{FA}(C)$ is the probability that negative samples are mapped to 1. Hence

$$\hat{p}_D = \frac{1}{n_{neg}} \sum_i X_i^{neg}$$

$$\hat{p}_{FA} = \frac{1}{n_{pos}} \sum_i X_i^{pos}$$

where $X_i^{neg}$, $X_i^{pos}$ are the labels of the positive and negative samples, respectively. These are each i.i.d. Bernoulli random variables.

The Hoeffding inequality can be directly applied:

$$Pr(|\hat{p}_D - p_D| \geq \epsilon) \leq 2 \exp\left(-2\epsilon^2 n_{neg}\right)$$

and similarly for $p_{FA}$. Now, we want a uniform bound that holds simultaneously for all possible threshold combinations. The number of these are $|\mathcal{T}_1||\mathcal{T}_2|$.

Define events $\mathcal{E}_{j,k}^D$ and $\mathcal{E}_{j,k}^{FA}$ to be the events that the estimated probabilities under the $j$th, $k$th threshold are within $\epsilon$, i.e.

$$\mathcal{E}_{j,k}^D := |\hat{p}_D(t_j^1, t_k^2) - p_D(t_j^1, t_k^2)| \leq \epsilon$$
$$\mathcal{E}_{j,k}^{FA} := |\hat{p}_{FA}(t_j^1, t_k^2) - p_{FA}(t_j^1, t_k^2)| \leq \epsilon.$$

Let's use the union bound across the $j, k$, we don't need it across FA/D, since these are independent.

$$Pr\left(\cap_{j=1}^{|\mathcal{T}_1|} \cap_{k=1}^{|\mathcal{T}_2|} \left(\mathcal{E}_{j,k}^D \cap \mathcal{E}_{j,k}^{FA}\right)\right) \geq \left(1 - 2|\mathcal{T}_1||\mathcal{T}_2| \exp\left(-2\epsilon^2 n_{neg}\right)\right)\left(1 - 2|\mathcal{T}_1||\mathcal{T}_2| \exp\left(-2\epsilon^2 n_{pos}\right)\right)$$

In other words,

$$\max_{j,k} \max(|\hat{p}_D(t_j^1, t_k^2) - p_D(t_j^1, t_k^2)|, |\hat{p}_{FA}(t_j^1, t_k^2) - p_{FA}(t_j^1, t_k^2)|) \leq \sqrt{\frac{\log(|\mathcal{T}_1|) + \log(|\mathcal{T}_2|)}{\min(n_{neg}, n_{pos})}},$$

with probability at least $\left(1 - \frac{2}{|\mathcal{T}_1||\mathcal{T}_2|}\right)^2$. Since this holds simultaneously for all threshold combinations, if we take the training-convex-hull area, the test AUC from using the frontier of that hull will be at most $2\epsilon$ smaller.

# B Kernel Mean Embedding View of Consistency Methods for Hallucination Detection

In what follows, we suppose that we have also access to the ground the truth $\pi^*(y|x)$ and observe $\frac{1}{n^*} \sum_{i=1}^{n^*} \delta_{y_i^*}, y_i^* \sim \pi^*(.|x)$.

**Proposition B.1.** *The optimal weight $\lambda$ for combining self and cross consistencies to approximates the consistency of the target with the ground truth satisfies:*

$$\min_{\lambda \in [0,1]} |\langle \mu_t, \lambda \mu_t + (1-\lambda)\mu_v \rangle - \langle \mu_t, \mu^* \rangle| \leq \min_{\lambda \in [0,1]} \sqrt{2(1 - \langle \mu_*, \lambda \mu_t + (1-\lambda)\mu_v \rangle)},$$

*Hence it is enough to solve:* $\max_{\lambda \in [0,1]} \langle \mu_*, \lambda \mu_t + (1-\lambda)\mu_v \rangle$. *Using an entropic regularization of this objective*

$$\max_{\omega_t, \omega_v \in [0,1], \omega_t + \omega_v = 1} \langle \mu_*, \omega_t \mu_t + \omega_v \mu_v \rangle - \varepsilon\left(\sum_{j \in \{t,v\}} \omega_j (\log \omega_j - 1)\right),$$

*we obtain* $\lambda^* = \frac{\exp(\frac{\langle \mu_t, \mu^* \rangle}{\varepsilon})}{\exp(\frac{\langle \mu_t, \mu^* \rangle}{\varepsilon}) + \exp(\frac{\langle \mu_v, \mu^* \rangle}{\varepsilon})}$.

$$|\langle \mu_t, \lambda\mu_t + (1-\lambda)\mu_v \rangle - \langle \mu_t, \mu^* \rangle| = |\langle \mu_t, \mu^* - (\lambda\mu_t + (1-\lambda)\mu_v) \rangle|$$
$$\le \|\mu_t\| \|\mu^* - (\lambda\mu_t + (1-\lambda)\mu_v)\|$$
$$\le \sqrt{2(1 - \langle \mu_*, \lambda\mu_t + (1-\lambda)\mu_v \rangle)},$$

last inequality follows from $\|\mu^*\| \le 1$, $\|\mu_t\| \le 1$, $\|\mu_v\| \le 1$ and $\|\lambda\mu_t + (1-\lambda)\mu_v)\| \le 1$ (since the kernel $\mathcal{E}$ is bounded by 1).

Hence to minimize this inequality it is enough to solve for :

$$\max_{\lambda \in [0,1]} \langle \mu_*, \lambda\mu_t + (1-\lambda)\mu_v \rangle$$

we can add to this an entropy regularizer

$$\max_{\omega_t, \omega_v \in [0,1], \omega_t + \omega_v = 1} \langle \mu_*, \omega_t\mu_t + \omega_v\mu_v \rangle - \varepsilon \Big( \sum_{j \in \{t,v\}} \omega_j(\log \omega_j - 1) \Big)$$

which gives us that: $\lambda = \omega_t = \frac{\exp(\frac{\langle \mu_t, \mu^* \rangle}{\varepsilon})}{\exp(\frac{\langle \mu_t, \mu^* \rangle}{\varepsilon}) + \exp(\frac{\langle \mu_v, \mu^* \rangle}{\varepsilon})}$ and $\omega_v = 1 - \lambda = \frac{\exp(\frac{\langle \mu_v, \mu^* \rangle}{\varepsilon})}{\exp(\frac{\langle \mu_t, \mu^* \rangle}{\varepsilon}) + \exp(\frac{\langle \mu_v, \mu^* \rangle}{\varepsilon})}$.

## C  Experimental Details

We conducted our experiments on eight RTX A6000 GPUs.

### C.1  Estimation of computation cost

To compute the budget, we use the estimation formula derived in Kaplan et al. (2020), Table 1, which states that the number of FLOPs per token is approximately $2N$ for models with sufficiently large dimensions, where $N$ represents the number of non-embedding parameters in the model. Then, for each question, the computation cost of cross-model consistency checking can be expressed as:

$$ml_q 2N_{verifier} + m^2 l_a 2N_{deberta}, \tag{2}$$

where $l_q$ is the number of tokens in the question, and $l_a$ is the number of tokens in each answer (assuming all $m$ answers share the same length), $N_{verifier}, N_{deberta}$ are the number of parameters of the verifier model and the entailment estimator (i.e., `deberta-v2-xlarge-mnli`), respectively. This estimation remains challenging due to the variability in the lengths of questions and answers, which depend on each specific question and model. To simplify this, we consider the scaling limit, where—assuming no restrictions—the lengths of questions and answers scale to the respective context lengths of the models processing them (since tokens exceeding the context length are dropped). Notably, the context length of `deberta-v2-xlarge-mnli` is only 512, while the context lengths of verifier models range from 4,096 to 128,000. Using this, we compute the ratio of the second term to the first term in Equation 2, which is $m\frac{l_a}{l_q}\frac{N_{deberta}}{N_{verifier}}$, in the limit where both lengths reach their maximum. We find that this ratio is no greater than 0.087 in the largest case in our settings, indicating a negligible contribution of the second term in the limit. Thus, we omit the second term to simplify the estimated additional computation per question to $ml_q 2N_{verifier}$. Similarly, we can derive the estimation for the cost of self-consistency checking as $ml_q 2N_{target}$. Based on this simplification, the metric *additional relative cost* can be estimated as $\frac{pN_{verifier}}{N_{target}}$, where $p$ accounts for the fact that the verifier model is queried for only a fraction $p$ of the questions in Algorithm 1.

### C.2  Additional Results

Figure 7 shows the comparison between the performance of existing methods and the approximated ceiling performance in terms of AURAC. We observe a pattern consistent with that in Figure 2.

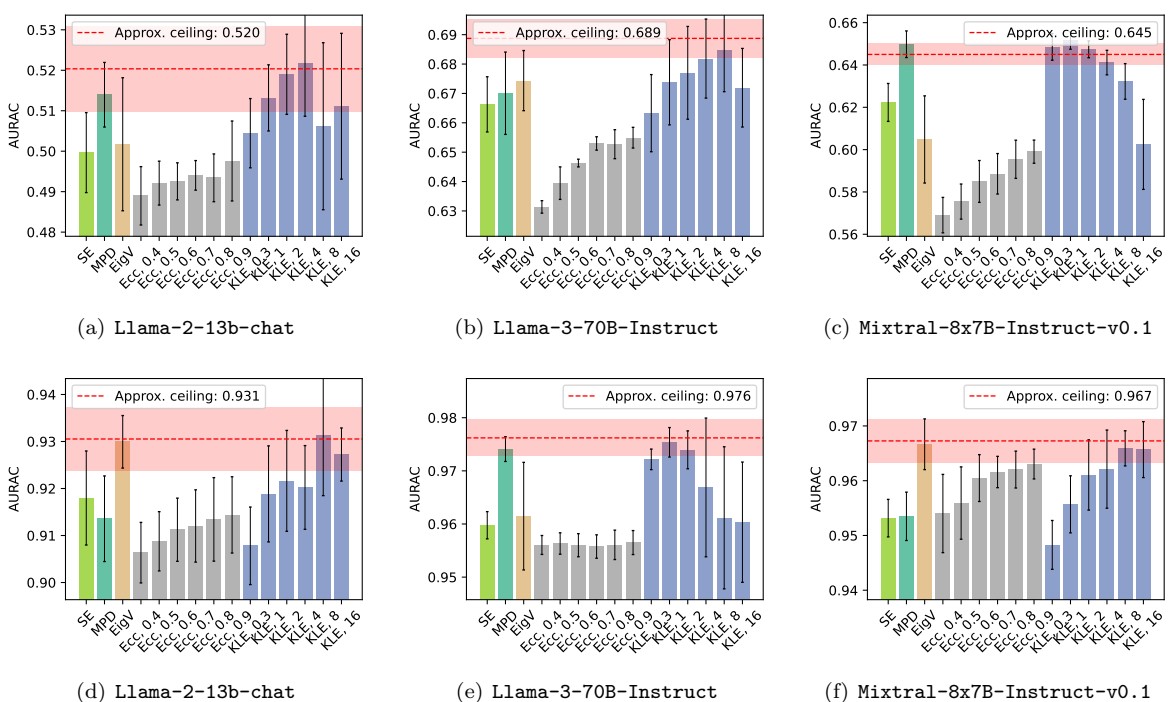

Figure 7: Comparison between the performance of existing methods and the approximated ceiling performance on the SQuAD ((a)–(c)) and TriviaQA ((d)–(f)) datasets. Here, performance is measured in terms of AURAC. We observe a pattern consistent with the discussion in Figure 2.

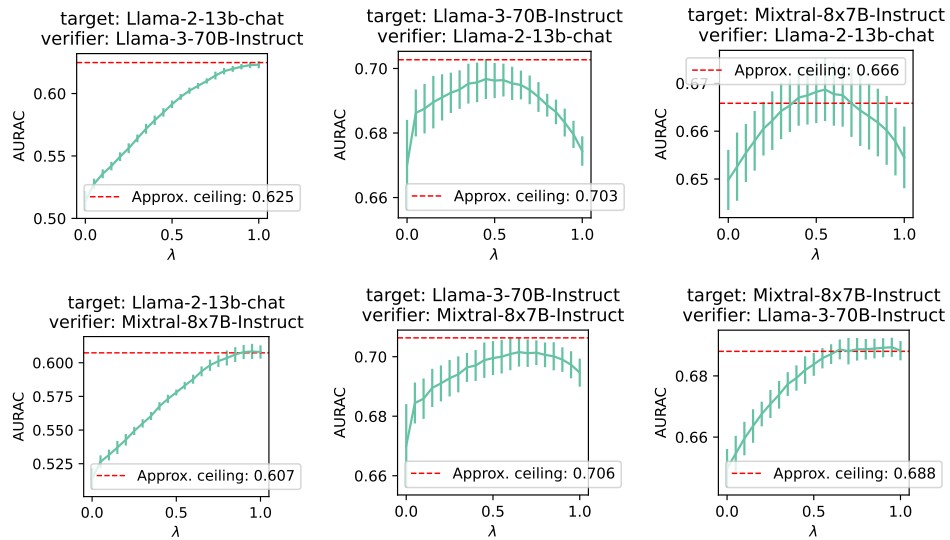

Figure 8: A simple weighted average of self-consistency and cross-consistency-based metrics, $(1 - \lambda)\text{MPD}(\boldsymbol{P}^{\text{self}}) + \lambda\text{MPD}(\boldsymbol{P}^{\text{cross}})$, can achieve performance close to that of the oracle method.

Figure 8 shows the AURAC performance against $\lambda$ when using a weighted average of self-consistency and cross-consistency-based metrics, $(1 - \lambda)\text{MPD}(\boldsymbol{P}^{\text{self}}) + \lambda\text{MPD}(\boldsymbol{P}^{\text{cross}})$.

Figure 9 shows the AUROC performance of the two-stage detection method under varying computational budgets.

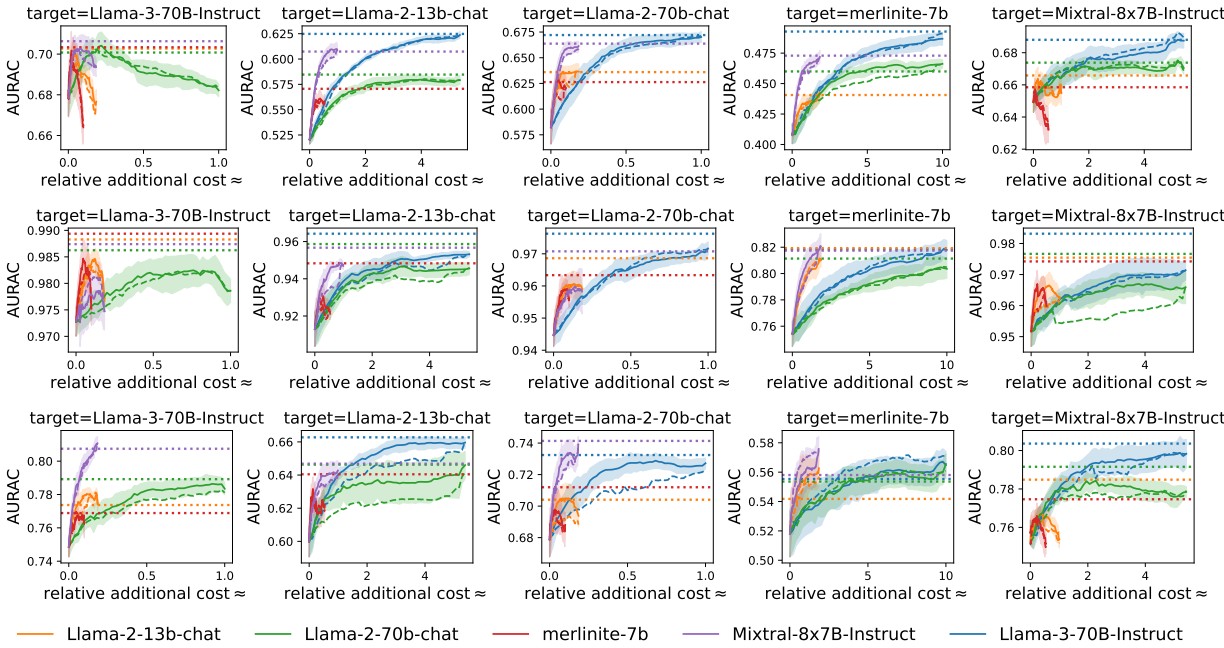

Figure 9: AURAC vs. relative additional cost for SQuAD (top), TriviaQA (middle), and Natural Questions (bottom).

We also provide an example where the weak verifier model's outputs are better than those of the target models. In this case, the self-consistency-based MPD is 0.183, while the cross-model-consistency-based MPD is 0.922.

**Question**: who is the original singer of fool if you think it's over

**Correct answer**: Chris Rea.

**10 answers from the target model (Llama-3-70B-Instruct):**

- Elkie Brooks is the original singer of "Fool (If You Think It's Over)".

- The original singer of "Fool (If You Think It's Over)" is Chris Rea.

- Elkie Brooks is the original singer of "Fool If You Think It's Over".

- Elkie Brooks is the original singer of "Fool If You Think It's Over".

- Elkie Brooks is the original singer of "Fool If You Think It's Over".

- Elkie Brooks is the original singer of "Fool If You Think It's Over".

- Elkie Brooks is the original singer of "Fool If You Think It's Over".

- Elkie Brooks is the original singer of "Fool If You Think It's Over".

- Elkie Brooks is the original singer of "Fool If You Think It's Over".

- Elkie Brooks is the original singer of "Fool If You Think It's Over".

**10 answers from the verifier model (Mixtral-8x7B-Instruct-v0.1):**

- The original singer of "Fool If You Think It's Over" is Chris Rea.

- The original singer of "Fool If You Think It's Over" is Chris Rea.

- The original singer of "Fool If You Think It's Over" is Chris Rea. He wrote and performed the song, which was released in 1978 as part of his album "Whatever Happened to Benny Santini?" The song gained popularity over the years and has been covered by various artists, including Elton John.

- The original singer of "Fool If You Think It's Over" is Chris Rea.

- The original singer of "Fool If You Think It's Over" is Chris Rea. This song is one of his most popular tracks and was released in 1978 as part of his album "Whatever Happened to Benny Santini?"

- The original singer of "Fool If You Think It's Over" is Chris Rea.

- Chris Rea

- The original singer of "Fool If You Think It's Over" is Chris Rea.

- The original singer of "Fool if You Think It's Over" is Chris Rea. He wrote and recorded the song for his 1978 album, "Whatever Happened to Benny Santini?". The song has since been covered by various artists, but Rea's version remains the most well-known and popular.

- The original singer of "Fool If You Think It's Over" is Chris Rea.

