# OpenReview forum: "Verify when Uncertain: Beyond Self-Consistency in Black Box Hallucination Detection"
_TMLR — Accepted by TMLR_

### Review · Reviewer_ZyB9 · 2025-11-28

**Summary Of Contributions:**

This paper explores the limitations of black-box self-consistency based methods for hallucination detection in LLMs and proposed a two-stage framework by employing a verifier model to conduct cross-consistency checks. This paper demonstrated improvements over self-consistency-based detection methods as well as analyzed the cost (overhead) of performing cross-consistency checks.

## Strengths:

- The authors have clearly demonstrated the limitations of self-consistency based detection methods which motivated this paper.
- The authors have carried out comprehensive evaluation to show the effectiveness of their proposed framework.

## Weaknesses:

- The explanation on the performance ceiling is a bit hand-wavy. Which model was used to estimate the entailment? It seems like Section 6 mentioned deberta-v2-xl-mnli but Section 4.1 mentioned deberta-large-mnli. Also, multiple figures showed that some metrics outperforming the ceiling, it would be nice if the authors can provide some clarification on that.
- Despite the theory, the novelty of the proposed Algorithm 1 is somewhat limited and recent works have adopted the idea of comparing outputs across different models for hallucination detection as well.

**Audience:**

Yes

**Audience Explanation:**

Hallucination detection on LLMs is an important issue of LLMs. This paper focuses on cross-consistency detection which has been the focus of many recent works.

**Claims And Evidence:**

Yes

**Claims Explanation:**

The claims are well-substantiated by empirical experiments. For empirical analysis, the paper compared five black-box detection baselines over various model families (Llama-2, Llama-3, and Mixtral) and datasets, and demonstrated consistent results.

The additional-cost analysis on the overhead of cross-consistency approaches is interesting.

**Requested Changes:**

- Some clarification and ablation studies on performance ceiling would secure my recommendation.

- Some visual fixes in the figures may be required (such as overlapping texts).

---

> ### Author Response · Authors · 2025-12-19
>
> We thank the reviewer for the helpful feedback and for acknowledging our contribution in demonstrating the limitations of self-consistency based detection and the comprehensiveness of our evaluation. Below, we address the reviewer’s questions.
>
> > Which model was used to estimate the entailment? It seems like Section 6 mentioned deberta-v2-xl-mnli but Section 4.1 mentioned deberta-large-mnli.
>
> The entailment estimator we used is deberta-v2-xl-mnli as stated in section 6. In Section 4.1 we mentioned deberta-large-mnli just as an example (we said “e.g., xxx”) but we never said we used it. We have changed this part to avoid any potential confusion.
>
> > Also, multiple figures showed that some metrics outperforming the ceiling, it would be nice if the authors can provide some clarification on that.
>
> Thank you for pointing this out. The “ceiling” we report is an approximated ceiling, obtained by training a GCN on finite samples to approximate the optimal mapping from consistency matrices to hallucination labels. In some cases, we observe that individual hand-designed metrics slightly outperform the approximated ceiling. This can arise due to finite-sample effects, limited model capacity of the GCN, and variance introduced by hyperparameter selection and validation splits. Importantly, these differences are small and fall within the reported standard deviations, and they do not change the overall conclusion that existing methods already operate very close to oracle performance.
>
> > Despite the theory, the novelty of the proposed Algorithm 1 is somewhat limited and recent works have adopted the idea of comparing outputs across different models for hallucination detection as well.
>
> We agree that comparing outputs across different models for hallucination detection has been explored in prior work, and we do not claim cross-model consistency as a new idea in isolation. Rather, our novelty lies in (1) a systematic analysis of the limits of self-consistency–based detection via an approximated ceiling; (2) the finding that cross-model consistency provides complementary information that can raise this ceiling, even when the verifier model is weaker than the target model; and (3) a principled, budget-aware two-stage algorithm that uses self-consistency to decide when cross-model verification is necessary, achieving significant performance gains while substantially reducing computational cost.
>
>
> > On the approximated ceiling
>
> We note that there is no direct way to measure the true ceiling performance; therefore, we use an empirical approximation, which is why we refer to it as an approximated ceiling in the paper. We believe that our GCN-based model provides a strong estimate of this ceiling. The model is trained in a supervised manner on a large in-distribution training set, and the two-layer GCN architecture is well matched to the input structure—matrices encoding pairwise entailment—which graph neural networks are particularly well suited to process. We also explored architectural and optimization variants: for example, adding a third layer slightly increased training time without improving performance, and using Adam rather than SGD led to significantly better results due to improved optimization stability. These experiments support our choice of model and hyperparameters as a reasonable configuration for extracting near-maximal predictive signals from the self-consistency matrix.
>
>
> > visual fixes in figures
>
> We fixed the overlapping issues in Figures 1 and 5. Please let us know if anything else needs to be improved.

---

### Review · Reviewer_w3Yg · 2025-11-29

**Summary Of Contributions:**

The paper addresses the saturation of self-consistency (SC) methods in black-box hallucination detection. It introduces two primary contributions: (1) An empirical demonstration using Graph Neural Networks (GCNs) that existing SC methods are near their theoretical performance ceiling; and (2) A budget-aware, two-stage algorithm that selectively employs a "verifier" LLM (cross-model consistency) only when the target model's SC score is uncertain.

**Audience:**

Yes

**Audience Explanation:**

The paper is explicit that it works in a black-box regime: no logits, no internal representations, only outputs and an external NLI model for entailment. This is consistently maintained in the methodology (Sections 3–4), and contrasts cleanly with white-box work cited in Related Work.

**Broader Impact Concerns:**

There are no concerns about ethical implications

**Claims And Evidence:**

Yes

**Claims Explanation:**

**A. Rigorous Quantification of the "Status Quo":**

The paper moves beyond "state-of-the-art chasing" by scientifically proving that current SC methods (MPD, EigenV, Semantic Entropy) are already operating near the information limit of the self-consistency matrix (Figure 2). This insight is valuable for the community, suggesting that future research should focus on external signals rather than refining SC aggregation metrics.

**B. Counter-Intuitive Findings on Verifier Strength:**

A strong empirical finding is that the verifier model does not necessarily need to be stronger than the target model to provide value. The experiments show that even a weaker verifier can improve detection performance (Section 4.2.1). This significantly enhances the practicality of the method, as users can deploy smaller, cheaper models (e.g., 7B parameter models) to verify larger models.

**C. Efficiency-Performance Convexity:**

The results in Figure 6 demonstrate that the performance-to-cost curve is convex. This implies that a large portion of the performance gain from cross-consistency can be unlocked with a disproportionately small computational investment (e.g., querying the verifier for only the most uncertain 10-20% of cases). This makes the method highly relevant for production environments where latency and token costs are constraints.

**D. Comprehensive Experimental Setup:**

The evaluation is robust, covering:

3 Datasets: TriviaQA, SQuAD, Natural Questions.

20 Model Combinations: A matrix of varying sizes (7B to 70B) for targets and verifiers.

Theoretical bounds: Comparison against the GCN-approximated ceiling.

**Requested Changes:**

**Ceiling is empirical, not a true upper bound**

- The performance “ceiling” is defined using a specific two-layer GCN oracle trained on Pself and ( Pself​,Pcross),

- This only bounds performance within that model class and dataset, not the true optimal function of these matrices.

- Claims about “little room for improvement” are therefore relative to this oracle, not information-theoretic limits.

**Narrow task scope: only QA-style hallucinations**

All experiments are on question-answering datasets (TriviaQA, SQuAD, Natural Questions).

- There is no evaluation on other important hallucination domains such as summarization, long-form generation, or multi-turn dialogue.

- It is unclear whether the conclusions about self-consistency saturation and cross-model gains generalize beyond QA.

**Little analysis of sampling and temperature hyperparameters**

- The approach depends on sampling 𝑚 high-temperature outputs from target and verifier models.

- The paper largely fixes 𝑚 and temperatures, citing prior work, without a detailed ablation in this specific setting.

- Practitioners get limited guidance on how performance and cost trade off as  𝑚 or temperature changes.

---

> ### Author Response · Authors · 2025-12-19
>
> We thank the reviewer for the constructive review and for finding our analysis of the state of the art rigorous, the empirical findings strong, the efficiency–performance trade-off compelling, and the experimental setup comprehensive. Below, we respond to the reviewer’s comments.
>
> > Ceiling is empirical, not a true upper bound. The performance “ceiling” is defined using a specific two-layer GCN oracle trained on Pself and ( Pself​,Pcross). This only bounds performance within that model class and dataset, not the true optimal function of these matrices.
> Claims about “little room for improvement” are therefore relative to this oracle, not information-theoretic limits.
>
>
> We agree that the ceiling we define is not an information-theoretic limit, but rather an empirical approximation. This is why we explicitly refer to it as an “**approximated ceiling**” in the paper to avoid any overstatement. At present, there is no known way to measure the true ceiling. However, our GCN-based model provides a strong estimate due to the following factors:
>
> (1) As the reviewer notes, the approximated ceiling is obtained by training a GCN model on a specific target model and dataset, and is therefore model- and data-dependent. If there existed a true optimal function that were model- and data-agnostic, then our approximated ceiling would only serve as an upper bound on this more general ceiling, since we optimize for maximal performance separately on each dataset and model. This only strengthens our conclusion: because existing methods already perform very close to our approximated ceiling—which may itself be higher than the true, model-agnostic ceiling—they must be at least as close, if not closer, to the true ceiling performance.
>
> (2) Supervision with oracle information. The GCN is trained using ground-truth hallucination labels, which constitute oracle information unavailable to other self-consistency-based methods. This access enables the model to directly learn a mapping from self-consistency matrices to hallucination labels.
>
> (3) Large, in-distribution training set. The training data is drawn from the same set of 400 questions used for evaluation, but with different randomly sampled responses. For each question, we generate 10 responses across 5 independent trials, yielding 2,000 training examples. This setup ensures exposure to diverse yet fully in-distribution data.
>
> (4) Architectural suitability and expressiveness. The two-layer GCN is well matched to the input structure: 10×10 entailment matrices that can be viewed as fully connected graphs with scalar edge weights and trivial node features (e.g., one-hot vectors). Given the small scale and simplicity of these graphs, the GCN has sufficient expressive capacity to approximate the function mapping entailment structure to ground-truth labels.
>
> Taken together, these factors suggest that our GCN-based approach can closely approximate the underlying ground-truth function, thereby providing a strong estimate of ceiling performance. While this is not a theoretical upper bound, we believe it serves as a meaningful and informative empirical approximation.

---

> > ### Author Response · Authors · 2025-12-19
> >
> > > Narrow task scope: only QA-style hallucinations …
> >
> >
> > We have revised the paper in several places in response to this comment.
> >
> > We agree that our experiments focus on QA-style hallucinations, and we have revised the manuscript (modified parts are highlighted in blue) to explicitly clarify this scope and avoid overgeneralization in the abstract and introduction.
> >
> > The three datasets we consider—TriviaQA, SQuAD, and Natural Questions—have been widely used in prior work on black-box hallucination detection, including Farquhar et al. (2024); Kuhn et al. (2023); Lin et al. (2023); and Nikitin et al. (2024). While recent benchmarks have begun to evaluate hallucinations in other generation tasks such as summarization and long-form generation, evaluation in these settings is generally more challenging, as it is harder to compare a model’s output with a ground-truth answer when both are long and open-ended. Determining whether two long passages are semantically consistent or whether a specific detail constitutes a hallucination is inherently ambiguous. Some evaluation, e.g., that of Manakul et al. (2023), even requires manual annotation of hallucinations, which can be expensive, time-consuming, and subjective. In contrast, QA has a more well-defined target. Therefore, we focus on QA-style benchmarks as a standard and controlled setting for studying consistency-based signals. We note that prior work such as Kuhn et al. (2023), Lin et al. (2023), and Nikitin et al. (2024) also evaluates only on QA-style tasks.
> >
> >
> > We also added the following discussion in Section 7: While our experiments are conducted on QA datasets, the proposed framework itself is task-agnostic. Both self-consistency and cross-model consistency operate purely on distributions of semantic entailment judgments. The ceiling analysis and the two-stage detection method do not rely on properties unique to QA, but instead on the distribution of entailment relations among sampled responses. As a result, we expect similar saturation phenomena and verifier-based gains to arise in other generation settings where semantic equivalence can be assessed.
> >
> >
> > > Little analysis of sampling and temperature hyperparameters. The approach depends on sampling 𝑚 high-temperature outputs from target and verifier models. The paper largely fixes 𝑚 and temperatures, citing prior work, without a detailed ablation in this specific setting. Practitioners get limited guidance on how performance and cost trade off as 𝑚 or temperature changes.
> >
> > The number of generations m has already been well studied in prior work on the same datasets we consider, and our consistency scores are built directly on these existing formulations. As a result, we do not re-study m separately. For example, Figure 2 in the supplementary material of Farquhar et al. (2024) and Figure 4 in Lin et al. (2023) both show that detection performance exhibits rapidly diminishing returns once m exceeds 5, with only marginal improvement from m=5 to m=10, and essentially no gain beyond that. Accordingly, in our work we follow the same choice used in prior studies (Farquhar et al., 2024; Kuhn et al., 2023; Lin et al., 2023; Nikitin et al., 2024).

---

### Review · Reviewer_JvBQ · 2025-12-08

**Summary Of Contributions:**

This paper investigates hallucination detection for black-box large language models and offers three main contributions that help clarify what consistency-based methods can and cannot achieve. First, the authors present a unifying perspective on a broad family of self-consistency–based detectors such as Semantic Entropy, MPD, KLE, and EigV. They show that all of these methods can be interpreted as functions operating over the same underlying entailment matrix constructed from diverse high-temperature generations. By training graph convolutional networks on these matrices, the authors approximate an upper bound, or “ceiling,” on how well any self-consistency method could perform. Their experiments reveal that current approaches already sit extremely close to this ceiling, leaving relatively little room for improvement within this paradigm.

The second contribution introduces cross-model consistency by incorporating a verifier LLM. Rather than relying exclusively on generations from the target model, the authors construct an expanded entailment matrix capturing relationships between target samples and verifier samples. This additional information substantially raises the performance ceiling, even when the verifier is weaker than the target model. Interestingly, the paper finds that a simple linear combination of self-MPD and cross-MPD nearly matches the oracle performance predicted by the GCN.

**Audience:**

Yes

**Audience Explanation:**

Hallucination detection is a problem of central importance to researchers and practitioners working with large language models, especially as black-box access becomes more common. The insights in this paper speak directly to ongoing questions in uncertainty estimation, model evaluation, and LLM reliability, topics that are highly relevant to TMLR’s readership.

**Broader Impact Concerns:**

The paper’s contributions generally have positive implications by helping reduce hallucinations in downstream systems, particularly in high-stakes applications where incorrect model outputs may have real-world consequences. However, several broader-impact considerations deserve more explicit discussion. One concern is that improved detectors may give users a false sense of security. Even with the proposed enhancements, hallucination detection remains imperfect, and over-reliance on these systems may encourage reductions in human oversight, see [1, 2].

References:

[1]. Gradual disempowerment: Systemic existential risks from incremental AI development

[2]. NATURAL EMERGENT MISALIGNMENT FROM REWARD HACKING IN PRODUCTION RL

**Claims And Evidence:**

Yes

**Claims Explanation:**

The central claims of the paper are that (1) self-consistency methods are already close to their theoretical performance ceiling, (2) cross-model consistency meaningfully raises this ceiling, and (3) a simple, cost-controlled two-stage approach achieves much of the possible performance improvement at a fraction of the computational cost. Across multiple datasets and a wide variety of model combinations, the authors provide consistent evidence supporting each of these statements. The ceiling analysis using GCNs is methodologically clear, and the comparisons between existing methods and the approximated oracle are easy to interpret. The experiments involving verifier models convincingly demonstrate the added value of cross-model information, and the ablations show that the improvements are robust even when the verifier is weaker than the target. Moreover, the cost-performance evaluations provide a realistic sense of how the method would perform under practical resource constraints. Although the work inevitably inherits some weaknesses from the use of GPT-4–generated labels, the empirical evidence itself is thorough, carefully presented, and aligns well with the authors’ claims.

**Requested Changes:**

The most important, in my opinion, is related to concerns about the use of GPT-4 for hallucination labeling. Since the entire evaluation pipeline depends on these labels, the paper would benefit from a more detailed discussion of the reliability of this labeling process, including possible biases or systematic errors. A small human comparison study or an analysis of known GPT-4 failure modes would help clarify how label noise might affect the estimation process.

Another clarification involves positioning the work in relation to prior cross-model approaches, such as SAC3, the distinction between prior contributions and the advancements presented here could be made more explicit.

---

> ### Author Response · Authors · 2025-12-19
>
> We thank the reviewer for the thoughtful review and for the positive assessment of our ceiling analysis and cross-model consistency results. Below, we address the reviewer’s concerns.
>
> > The most important, in my opinion, is related to concerns about the use of GPT-4 for hallucination labeling …
>
> Using GPT-based annotation is consistent with prior work (Manakul et al., 2023; Farquhar et al., 2024; Kuhn et al., 2023; Lin et al., 2023; Nikitin et al., 2024). In addition, an analysis of GPT-4’s use for labeling hallucinations, compared with human annotators, is provided in Farquhar et al. (2024). Specifically, in the supplementary material (“Note 6: Assessing Model Accuracy”), the authors report that two human raters agreed with each other at roughly the same rate (92%) as they agreed with GPT-4 on average (93%). This suggests that GPT-4’s evaluations are, on average, reasonably close to those of human raters. We have added this information as a footnote in Section 6.
>
>
> > Another clarification involves positioning the work in relation to prior cross-model approaches, such as SAC3, the distinction between prior contributions and the advancements presented here could be made more explicit.
>
> We clarify the distinction from prior work on cross-model approaches in three key ways. First, we systematically study different combinations of target and verifier models with varying capabilities, whereas prior work such as SAC3 considers only a single target model with two verifier models. This demonstrates that the benefits of cross-model consistency hold broadly across various types of target-verifier pairs and are not tied to a specific model pairing. Second, we show that incorporating cross-model consistency can raise the performance ceiling, providing principled evidence supporting its benefit, namely, that even when self-consistency information is used to its maximal extent, adding cross-model consistency remains beneficial. Third, we propose a budget-aware two-stage algorithm, together with a theoretical interpretation, that queries the verifier model for only a controllable fraction of inputs, enabling practical control over computational cost while retaining most of the performance gains.

---

### Decision · Action_Editor_ZjDc · 2026-06-15

**Recommendation:** Accept as is

**Audience:**

Yes

**Audience Explanation:**

Using test-time compute for improving performance is a very important topic now in AI and the analysis will be interesting to many. The method as well might be use by some for lightweight improvements in performance.

**Claims And Evidence:**

Yes

**Claims Explanation:**

The authors present an interesting analysis on the ceiling of using self-consistency showing we are near that and a way to use even weaker models with cross-consistency to raise that ceiling, the analysis and experiments are carried out well.